# Clonal and Scalable Endothelial Progenitor Cell Lines from Human Pluripotent Stem Cells

**DOI:** 10.3390/biomedicines11102777

**Published:** 2023-10-13

**Authors:** Jieun Lee, Hal Sternberg, Paola A. Bignone, James Murai, Nafees N. Malik, Michael D. West, Dana Larocca

**Affiliations:** 1AgeX Therapeutics, Inc., 1101 Marina Village Parkway, Alameda, CA 94501, USA; hsternberg@agexinc.com (H.S.); pabignone@gmail.com (P.A.B.); nmalik@agexinc.com (N.N.M.); danaclarocca@gmail.com (D.L.); 2Advanced Cell Technology, Alameda, CA 94502, USA; 3Reverse Bioengineering, Inc., Alameda, CA 94502, USA; mwest@reversebio.com

**Keywords:** endothelial progenitor cells, ischemic disease, cardiovascular diseases, exosomes

## Abstract

Human pluripotent stem cells (hPSCs) can be used as a renewable source of endothelial cells for treating cardiovascular disease and other ischemic conditions. Here, we present the derivation and characterization of a panel of distinct clonal embryonic endothelial progenitor cells (eEPCs) lines that were differentiated from human embryonic stem cells (hESCs). The hESC line, ESI-017, was first partially differentiated to produce candidate cultures from which eEPCs were cloned. Endothelial cell identity was assessed by transcriptomic analysis, cell surface marker expression, immunocytochemical marker analysis, and functional analysis of cells and exosomes using vascular network forming assays. The transcriptome of the eEPC lines was compared to various adult endothelial lines as well as various non-endothelial cells including both adult and embryonic origins. This resulted in a variety of distinct cell lines with functional properties of endothelial cells and strong transcriptomic similarity to adult endothelial primary cell lines. The eEPC lines, however, were distinguished from adult endothelium by their novel pattern of embryonic gene expression. We demonstrated eEPC line scalability of up to 80 population doublings (pd) and stable long-term expansion of over 50 pd with stable angiogenic properties at late passage. Taken together, these data support the finding that hESC-derived clonal eEPC lines are a potential source of scalable therapeutic cells and cell products for treating cardiovascular disease. These eEPC lines offer a highly promising resource for the development of further preclinical studies aimed at therapeutic interventions.

## 1. Introduction

Cardiovascular diseases including atherosclerosis, ischemic stroke, myocardial infarction, and ischemic cardiomyopathy currently represent a large economic and societal burden due to their high mortality and disability rates [1]. Impairment of blood vessel formation and maintenance is one of the major underlying causes of many age-associated diseases such as diabetes, cardiovascular disease, peripheral artery disease, and slow wound healing [2]. Aging of angiogenic progenitor populations in adult tissues may account for the lack of homeostatic repair and maintenance of vascular tissues seen with advanced age [3]. Regenerative cell replacement therapies offer a potentially effective strategy for treating conditions involving dysfunctional endothelium. Adult stem cell therapies using mesenchymal stem cells (MSCs) [4,5], endothelial progenitor cells (EPCs) [6], and cardiosphere-derived cells (CDCs) [7] are currently being investigated as potential therapeutic agents for ischemic diseases [8]. However, the potential of adult stem cell therapies is challenged by issues of identity, scalability, stability, and purity. Human pluripotent stem cells (hPSCs) have the potential to provide a scalable source of nearly all human somatic cell types [9]. Multiple groups have explored the therapeutic efficacy of hPSC-derived endothelial cells (hPSC-ECs) in animal models of ischemic cardiovascular diseases and demonstrated a potential for enhancing angiogenesis, tissue perfusion, and organ graft [10,11,12]. However, current reports of vascular cell differentiation from hPSC show results in inefficiency (1–5%) [13,14], heterogeneous aggregates [15] or lack of consistent yields of endothelial cells [16]. Manufacturing hPSC-derived cells at scale presents a problem because of the difficulty reproducing the differentiation protocol for each batch and issues with cost-effective scaling of hPSCs [17]. Current hPSC cell-based therapies in clinical testing such as those being investigated for treating macular degeneration (RPE cells) and spinal cord injury (Oligodendrocytes) require relatively small doses or small patient populations [17,18]. In contrast, cardiovascular disease treatments may necessitate larger doses, and hence more stringent release criteria. Given the heterogeneous nature of hPSC-ECs and their various differentiation protocols, clinical application in disease treatment remains challenging [19,20].

We have previously developed a method of generating purified and embryonic site-specific somatic cell types through the propagation of hPSC-derived clonal human embryonic progenitor cell lines to overcome current issues of purity and scale [21]. We designated these cultures as human “embryonic progenitor” (hEP) cells because of their ability to self-renew under selected culture conditions, their persistent expression of embryonic developmental stage gene markers such as *PCDHB2*, and their lack of fetal/adult gene markers such as *COX7A1* that are preferentially expressed in cells that have traversed the embryonic-fetal transition [22]. These hEP cell lines also typically display limited lineage potential based on the loss pluripotency markers and pluripotent functionality. In our initial characterization of approximately 200 hEP lines, we reported that they were often capable of robust expansion and displayed a diversity of >140-fold distinct cell types [21]. Due to the diversity and clonal nature of hEP cell lines, the cells show site-specific markers such as homeobox genes that facilitate the identification of the lines as precursors to specific embryonic anlagen. For example, we characterized adipocyte progenitor cells [21,23], seven distinct osteochondral progenitor cell types [24,25], and progenitors of cranial neural crest [26] from our library of hEP cell lines. These hEP cell lines derived from hPSCs were partially differentiated and selected for scalable clones. More importantly, they were able to further differentiate into (1) brown adipocyte cells for metabolic disease treatment, (2) chondrocytes for cartilage regeneration, and (3) cellular components of the choroid plexus, all of which have unique progenitor properties for cell-based therapy [21,22,23].

In the present study, we derived novel hEP cell lines with endothelial properties and characterized a panel of 14 embryonic endothelial progenitor cell (eEPC) lines to explore their potential as a scalable source of human endothelial cells and cell products for further therapeutic development. We compared the eEPC lines to a diverse panel of adult endothelial cell (AEC) lines using comprehensive transcriptomic analysis. We found that 13 out of 14 eEPC lines expressed a broad array of endothelial specific genes shared across various endothelial subtypes. We further assessed these eEPC lines for endothelial identity using cell surface markers and well-established vascular tube forming assays. Interestingly, the eEPCs were clearly distinguished from AECs based on embryonic specific gene expression and molecular network analysis. In addition, we were able to achieve high scalability typical of previously identified hEP cell lines and stable secretion of angiogenic exosomes, suggesting that eEPCs may be advantageous for developmental research and for treating cardiovascular disease in a large patient population.

## 2. Materials and Methods

### 2.1. Derivation of Endothelial Progenitor Cell Lines

Research grade hESCs (ESI-017) were differentiated using a two-step progenitor derivation protocol. In the first step, human embryonic stem cells were exposed to a 7-day procedure on Matrigel coated plates involving specific differentiation factors, followed by exposure to two different growth media to generate two cultures of heterogeneous cells designated as Candidate Cultures (CC). Afterwards, each CC subpopulation was placed at clonal densities in their same growth medium for two weeks. Then, individual clonal colonies were selected with cloning cylinders and expanded from a 24-well plate to multiple T225 flasks (Figure 1A).

As described above and in Figure 1 [13], hESCs were differentiated to promote mesoendoderm/endoderm commitment in the presence of specific growth factors for specific periods of time to obtain the desired endothelial progenitor characteristics. Briefly, after reaching confluence, hESCs were exposed to: (day 0), basal differentiation medium supplemented with 25 ng/mL WNT3A (removed at day 1); and 100 ng/mL activin A (removed at day 3); on day 3, medium was supplemented with 30 ng/mL FGF-4 and 20 ng/mL BMP-2 (remained for the duration of culture). On day 7, the differentiated cells were divided into two growth media as: (1) Microvascular endothelial growth medium (EGM-MV2, Cat# C-39226, PromoCell, GmbH, Heidelberg, Germany) with added (TGF) β -inhibitory molecule SB431542 (10 μM) and (2) Smooth Muscle Growth Medium (SM2) (PromoCell, Cat# C22062) with added SB431542 (10 μM). After further expansion in the two growth media, the cells were seeded at clonal dilution for two weeks. Individual clonal colonies were then selected and placed in a 24-well plate (each clonal colony/well given an independent name) and scaled to multiple T225 flasks before they were frozen down and later thawed for further analysis as described in West et al. [21] (Figure 1B and Appendix A). Throughout the expansion and analysis of eEPCs, cells were continually maintained in the presence of SB431542 in their original growth medium that they were exposed to after the 7 days differentiation procedure.

### 2.2. Cell Lines and Culture

A total of 14 eEPC lines and 15 human adult endothelial cell (AEC) lines were cultured in endothelial growth medium (EGM-MV2, Cat# C-39226, Lot# 25 x 461M089, PromoCell, GmbH, Heidelberg, Germany) on gelatin-coated plates. All the information on cell lines including cell origins and sources were described in Appendix A. The medium was changed every 2–3 days and cells were passaged at 80% confluency. The cells when maintained in the undifferentiated state were cultured at 37 °C in a humidified atmosphere of 10% CO_2_ and 5% O_2_.

### 2.3. Transcriptomic RNA-Sequencing Analysis

RNA was prepared upon lysis with RLT with 1% β-mercaptoethanol, using Qiagen RNeasy mini kits (Cat# 74104) following manufacturer’s directions. The extracted RNA was then quantitated using a NanoDrop (ND-1000) spectrophotometer. Library Construction was performed using the Illumina Truseq mRNA library prep kit following manufacturer’s directions. Library QC and library pooling were accomplished using an Agilent Technologies 2100 Bioanalyzer to assay the library fragments. qPCR was used to quantify the libraries. Libraries were pooled, which had different barcodes/indexing and sequencing, in one lane. The paired-end sequencing was performed using the Illumina HiSeq4000 sequencing instrument, yielding 100-bp paired-end reads. The sequencing was performed by BGI Americas Corporation. Data analysis of the transcription levels (FPKM values) was performed using Gene Spring GX Suite (Agilent, Santa Clara, CA, USA). Hierarchical and correlation analysis were performed with a moderate *t*-test and p (Corr) cut-off = 0.05 while applying Pearson metric. The dendrogram was created using hierarchical cluster analysis using the average agglomeration method. For each cell culture sample, the differential expression values were calculated against a selection of 14 eEPC clones versus 15 adult endothelial cell lines versus 17 non-endothelial cell lines including embryonic and adult origin. All the information on samples is described in Appendix A. Gene ontology analysis was performed using the PANTHER Test from the GO Consortium with FDR correction. Raw data used for the current study are available from the corresponding author upon request.

### 2.4. Whole Genome Microarray Analysis

Total DNA was extracted from cells using Qiagen DNeasy mini kits according to instructions supplied by the manufacturer. DNA concentrations were measured using a Nanodrop spectrophotometer. Whole-genome expression was obtained using Illumina Human HT-12 v4 BeadArrays. In preparation for Illumina BeadArrays, total DNA was linearly amplified and biotin-labeled using Illumina TotalPrep kits (Life Technologies, Temecula, CA, USA). The sample quality was measured using an Agilent 2100 Bioanalyzer before being hybridized to Illumina BeadChips, processed, and read by an iScan microarray scanner according to the manufacturer’s instructions (Illumina, San Diego, CA, USA). Values under 130 relative fluorescence units (RFUs) were considered as nonspecific background signal. Analysis of microarray data was performed using the GeneSpring suite. Raw microarray data were normalized with the R BeadArray library [27]. Merging of data from different experiments and their subsequent quantile normalization was performed using functions combine and lumiN, respectively, of the lumi library. Dendrograms were created by hierarchical cluster analysis, correlation and 3-dimensional PCA analysis were created in the GeneSpring suite.

### 2.5. Statistical Analyses

All data are expressed as mean ± standard deviation. Statistical analysis between groups at each time point was performed by the unpaired student’s *t*-test. Independent experiments of samples over time were analyzed by repeated measures of ANOVA with the Holm adjustment. Differences were considered significant at probability values of *p* < 0.05.

### 2.6. Flow Cytometry

Cells were dissociated into a single-cell suspension by using TrypLE (Invitrogen-Life Technologies, Waltham, MA, USA) and fixed in BD Cytofix buffer (BD Biosciences, Franklin Lakes, NJ, USA) for 20 min at room temperature. The cells were permeabilized by washing and incubating them with BD Permeabilization/Wash (BD Biosciences, Franklin Lakes, NJ, USA) buffer at 1 × 10^6^ cells per 1 mL for 10 min. The cells were stained by incubating them with monoclonal antibodies (anti-human CD34-APC and anti-human CD31-FITC, ThermoFisher Scientific, Waltham, MA, USA) for 30 min. Primary antibodies were diluted according to the manufacturer’s instructions. Mouse IgG1 kappa Isotype Control (P3.6.2.8.1), Mouse IgG1 Isotype-FITC were used as controls. The cells were scanned with an LSRII flow cytometer (BD Biosciences, Franklin Lakes, NJ, USA) and analyzed with the FlowJo software version 10.6 (Ashland, OR, USA).

### 2.7. Immunocytochemistry

For detection of CXCR4 and CD31, cells were washed once with PBS and fixed in 4% paraformaldehyde for 30–60 min at room temperature (RT). Fixed cells were washed three times with PBS, permeabilized, and blocked by incubation in blocking buffer (5% normal donkey serum, 1% BSA, and 0.1% Triton X-100 in PBS) for 1 h at RT. The cells were then incubated overnight at 4 °C with CD31 monoclonal antibody with Alexa Fluor 488 (Thermo Sci. MA5-18135, Waltham, MA, USA) and primary rabbit anti-human CXCR4 polyclonal antibody (Thermo Sci. PA1-24894, Waltham, MA, USA) at a dilution of 1:500 in 5% normal donkey serum, 0.5% BSA, and 0.05% Triton X-100 in PBS. Then, the cells were washed four times with PBS plus 0.05% Triton X-100 (PBS-Triton) and incubated for 1 h at RT with Alexa Fluor 568 donkey anti-rabbit IgG antibody (Invitrogen, A10042, Waltham, MA, USA) at a 1:500 dilution in PBS-Triton. Isotype controls were stained under identical conditions except that total rabbit IgG (Life Technologies, 10500C, Carlsbad, CA, USA) was used as primary antibody. Cells were counterstained with DAPI at 0.1 ng/mL for 10 min at RT and imaged on a Nikon Eclipse TE2000-U inverted microscope.

### 2.8. Vascular Tube Forming Assay

Tube formation angiogenesis assay was performed using the Cell Player Angiogenesis PrimeKit (Essen Bioscience, Ann Arbor, MI, USA) to assess tube network growth in eEPC lines. eEPCs were labeled using TagRFP control vector, which consists of the lentiviral backbone vector, pLKO.1-puro, containing a gene encoding red fluorescence protein (RFP), driven by the CMV promoter (Millipore Sigma, Cat#SHC012, Burlington, MA, USA). RFP labeled eEPCs were plated with monolayer, and 10ng/mL of recombinant human VEGF protein (R&D System, Cat# 293-VE-010, Minneapolis, MN, USA) was treated to enhance tube formation. Images were taken every 6 h for 10–14 days using an IncuCyte imager. Tube formation was quantified using the IncuCtye Angiogenesis Analysis Module by dividing the lengths of all tube networks by the image area at each time point (mm/mm^2^). Exosomes for tube formation assay were isolated from eEPC and BM-MSC (PromoCell, Heidelberg, Germany) culture medium using the precipitation method following the manufacturer’s instructions (Invitrogen, Cat#4478359, Waltham, MA, USA). Total exosome particles were counted using nanoparticle tracking analysis (NTA).

Tube formation angiogenesis assay was also performed using human umbilical vein cells (HUVECs) that were seeded onto growth factor reduced Matrigel^®^ (Corning #354230, Corning, NY, USA) to assess tube network growth effect of eEPCs-secreted exosomes. HUVECs with 10^4^ cells/well were seeded in µ-Slide 15 well 3D (ibidi, Cat #81506, Gräfelfing, Germany) coated with Matrigel and followed by the treatment of eEPC-exosomes in PBS, compared with control conditions (e.g., complete growth media, EGM-MV2, and PBS only). Tube formation networks were analyzed using the ImageJ version 1.44 Angiogenesis Analyzer.

## 3. Results

### 3.1. Derivation of Clonal Embryonic Endothelial Progenitor Cell (eEPC) Lines

We have previously demonstrated that a large variety of distinct, highly scalable, and clonally pure progenitors can be isolated from human pluripotent stem cells using a two-step process of differentiation under diverse conditions followed by clonal selection of scalable cell lines [21]. In our initial clonal derivation, we isolated over 200 clonal cell lines and found by non-negative matrix factorization that 140 of these were distinct cell types [21]. In our original study, we randomly generated a high level of progenitor cell diversity using various initial differentiation conditions followed by clonal selection for producing scalable, robust cell lines. For deriving endothelial cell progenitors, we instead exposed human embryonic stem cells (hESCs) to specific directed differentiation conditions, followed by expansion and clonal selection with further expansion to obtain scalable endothelial progenitor lines. We used both endothelial (RP1) [13] and endoderm (protocol “30”) lineage-directed differentiation conditions (Figure 1 and Appendix A). The cell morphology suggested that the endodermal differentiation conditions (protocol “30”) resulted in clonal endothelial lines, which was later confirmed by transcriptomic analysis (Figure 2). The cell lines established by the RP1 method, in contrast, resulted in flat elongated cells, as we previously reported [28]. Whole genome expression analysis comparing clones from RP1 differentiation protocol to clones from “30” differentiation protocol revealed a significant difference in gene expression profile. The cell lines from RP1 method clustered with primary pericyte cell lines (Appendix A) [28]. In the current study, we focus on establishing endothelial progenitor cell lines using protocol “30”. In Stage 1 of protocol “30” differentiation, we induced mesenchymal to endoderm differentiation with Activin A and Wnt3A treatment for 3 days. Afterwards, in Stage 2, FGF-4 and BMP-2 were used to generate large quantities of eEPCs. Based on this differentiation system, we harvested the intermediate progenitor cells according to surface antigen expression and performed RNA-sequencing analysis. Clonal lines that were scalable in their selective medium were cultured in progressively larger flasks up to roller bottle culture, then harvested and banked.

### 3.2. Transcriptomic Analysis Indicates the Clonal eEPC Lines Are Endothelial Cells

To understand the molecular characteristics of the clonally selected hEP lines, we analyzed whole transcriptomic gene expression. We compared three groups: (1) 14 human embryonic endothelial progenitor cell (eEPC) lines, 30-MV2-6, 30-MV2-3, 30-MV2-4, 30-MV2-10, 30-MV2-17, 30-MV2-19, 30-MV2-2, 30-MV2-7, 30-MV2-9, 30-MV2-14, 30-MV2-24, 30-MV2-8, 30-SM2-1, and 30-SM2-3; (2) 15 primary adult endothelial cell (AEC) lines including HBMEC, HCPEC, HDMEC, HDLEC, HLEC, HIMEC, HRGEC, HHSEC, HCMEC, HAEC, HOMEC, HUVEC, HUAEC, HMVEC, and HAEC; and (3) a panel of 17 non-endothelial cell lines (both embryonic and fetal/adult). The description of cell origin, cell sources, and ages of RNAseq samples are described in Appendix A. Pearson correlation analysis, principal component analysis (PCA), and unsupervised clustering of whole transcriptomic data were performed, and the comprehensive analysis indicated that 13 out of 14 eEPC lines clustered with the primary AEC lines, demonstrating a strong correlation between eEPC and AEC lines, whereas both adult and embryonic non-endothelial cell lines were distinct from eEPC and primary AEC lines (Figure 2 and Appendix A).

Hierarchical clustering analysis with a representative heatmap in Figure 3 showed that 13 out of 14 eEPC lines shared a group of endothelial specific genes with primary AEC lines. Pathway enrichment analyses of 399 genes that were commonly expressed in both eEPC and AEC lines was performed (Figure 3A). The biological gene ontology (GO) terms showed that endothelial cell differentiation (GO 0045446), vascular development (GO 0048514, GO 0001944, and GO0001568), and angiogenesis (GO 0001525) were significantly enriched (Figure 3B), suggesting that 13 clonal eEPC lines express endothelial characteristics comparable to primary endothelial cells.

### 3.3. eEPC Lines Express Surface Proteins and Cellular Markers of Endothelial Cells

To further assess the endothelial characteristics of eEPC lines, we initially tested the expression of the key endothelial markers, such as cluster of differentiation (CD) 31, VE-Cadherin (CD144), vascular endothelial growth factor receptor (VEGFR-2)/KDR, and Von Willebrand factor (vWF), which are all widely recognized as endothelial-specific markers [10,29] and CD34, which is known as an EPC marker [30]. Phase contrast imaging of representative eEPC lines displayed a cobblestone shape morphology (Figure 4A), and 98.9% eEPC population expressed CD34+/CD31+ cells according to single cell flow cytometric analysis (Figure 4B). We also confirmed the CD31 expression in a representative eEPC line, 30MV2-6, using immunofluorescence staining (Figure 4C) as well as transcriptomic gene expression of endothelial-specific genes (*KDR* and *vWF*) (Appendix A). Interestingly, we observed that the eEPC lines expressed platelet-derived growth factor-B (*PDGF-B*), inhibitor of differentiation 1 (*ID1*), and C-X-C Motif Chemokine Receptor 4 (*CXCR4*), with comparable level to AEC lines (Appendix A). It has been reported that *PDGF-B*, *ID1*, and *CXCR4* play a role in stimulating vascular circulation [31], preserving vascular commitment via TGFβ inhibition [13], and regulating tissue regeneration as well as angiogenesis [32], respectively. These data suggest that the eEPC lines possess global gene expression profiles regulating a molecular network of endothelial cells.

### 3.4. eEPC Lines Enhance Tube Formation in Response to VEGF

We next investigated the angiogenic functionality of eEPC lines for vascular tube formation *in vitro*. To assess tube network growth in eEPC lines, eEPCs were labeled using TagRFP and co-cultured with normal human dermal fibroblasts (NHDF). This model allows us to demonstrate all phases of the angiogenesis process, including proliferation, migration, and, eventually, differentiation and angiogenic networks [33]. Imaging the co-culture in live-cell analysis system enabled us to identify RFP-labeled eEPCs from co-cultured NHDF and to visualize the vessel formation networks of eEPC over 6 days (Figure 4D and Appendix A). To quantify the amount of tube formation, we combined time-lapse image acquisition to measure network tube length. Figure 4E shows tube formation that is responsive to VEGF, an integral proangiogenic cytokine, in two representative eEPC lines (30MV2-4 and 30MV2-14) over 6 days. The angiogenic potency and response to VEGF further supports the endothelial nature of these eEPC lines.

### 3.5. eEPC Lines Retain Embryonic Phenotype

To better understand the molecular signature of eEPC lines compared to AEC lines, we performed a differential gene expression analysis of the transcriptomic RNAseq data. Although the overall global gene expression patterns of the 13 eEPC lines were similar to 15 AEC lines from various origins, we found that there was a significant subset of genes that were differentially expressed in the eEPC lines (Figure 5). The RNAseq data were validated by the independent measurement of showing global gene expression profile using microarray analysis (Appendix A). To define a specific identity of eEPC lines, differentially expressed genes in eEPCs were compared to genes expressed in AECs and ranked according to the average fold change in transcript expression. Indeed, we observed 1092 genes with at least 2-fold increased expression and 579 genes with at least 2-fold decreased expression in eEPC lines relative to AEC (Figure 6A). Pathway enrichment analysis showed that eEPC-enriched genes were related to vascular cell migration, vascular transport, and vessel development (Figure 6B). In contrast, AEC-enriched gene pathways related to regulation of mitochondrial ATP synthesis, mitotic spindle elongation, and metabolic processes. We found a set of genes reported as major endothelial regulators that were highly upregulated in eEPC lines, including apelin receptor (*APLNR*), differentiation marker of hematopoietic stem and progenitor cells [34], insulin-like growth factor 2 (*IGF2*) promoting EPC homing [35], *MAFB*, controlling endothelial sprouting [36], insulin-like growth factor binding protein 5 (*IGFBP5*), promoting angiogenesis [37], *H19*, a long non-coding RNA increasing EC tube formation [38], Carnitine palmitoyltransferase 1 C (*CPT1C*) promoting human mesenchymal stem cells survival [39], Tartrate-resistant acid phosphatase 5 (*ACP5*), promoting cell differentiation and proliferation [40], and enhanced prolyl hydroxylase domain (PHD)-3 (*EGLN3*), a regulator associated with resistance to stress) [41] (Figure 7A and Appendix A).

We previously identified *COX7A1*, encoding a cytochrome C oxidase subunit, as a novel marker associated with the mammalian embryonic-fetal transition (EFT) that is almost exclusively expressed in fetal and adult cells [22]. *COX7A1* expression level in eEPC lines was observed to be undetectable (Log2 FC = 0.1), compared to AEC lines (Log2 FC = 4.3), suggesting that eEPC lines retained an embryonic progenitor stage (Figure 7B). In addition, many genes were AEC specific and low or undetectable in eEPC lines. For example, expression of CC chemokine ligand 2 (*CCL2*), which is associated with diabetes [42], β-site APP cleavage enzyme 2 *(BACE2*), which is linked to increased risk and earlier disease onset such as Alzheimer’s disease [43], an antioxidant enzyme (*CAT*), which plays an important role in endothelial cell shear stress response [44], chemokine (C-X-C motif) ligand 1 (*CXCL1*), which regulates TNF-stimulated endothelial cells [45], interferon α-inducible protein 27 (*IFI27*), which is a stimulator of VEGF-A mediated angiogenesis [46], matrix la protein (*MGP*), which regulates differentiation of endothelial cells [47], and a long non-coding RNA that regulates angiogenesis (*MEG3*) [48], were all down regulated in eEPC relative to AEC (Figure 7B and Appendix A). Taken together, we identified a subset of genes that were differentially expressed in eEPCs compared to AECs, suggesting that eEPCs, while clearly expressing an endothelial gene network, have not undergone terminal endothelial differentiation to fetal/adult but instead express an embryonic pattern of gene expression.

### 3.6. Stable Production of Angiogenic Embryonic Endothelial Progenitor Cells (eEPCs)

We examined cell growth characteristics by measuring scalability and functional stability of a representative eEPC line, 30-MV2-6. It has been suggested that adult EPCs have therapeutic potential and clinical implications in ischemic diseases [6,49,50,51]. Therefore, the stable and scalable production of eEPC lines may be beneficial for supporting preclinical and clinical research on cell-based therapy.

To test cell production capacity of eEPC lines, cumulative population doublings (pd) of cells was compared to primary MSCs. We found that MSCs begin to senesce and lose proliferative and differentiation capacity after 12 pd, consistent with previous reports [52,53]. In contrast, the eEPC line showed an exponential growth up to at least 80 pd (Figure 8A). To assess the angiogenic activity of eEPCs, we tested the secreted exosomes from eEPC conditioned medium in a human umbilical vein endothelial cell (HUVEC) vascular tube formation assay. The conditioned medium from a representative eEPC line, 30-MV2-6, was precipitated to isolate exosomes. Addition of eEPCs exosomes resulted in enhanced HUVEC tube formation compared to basal medium alone and was equivalent to complete endothelial cell growth medium (CGM) containing VEGF, IGF and FGF (Figure 8B). These data suggest that exosomes secreted from eEPCs contain angiogenic factors. Recent studies have shown that exosomes are secreted from a variety of cell types including MSCs. They play a role in intercellular communication through the transfer of their cargo carrying lipids, proteins, and RNAs to recipient cells [54]. In this study, we initially observed that exosomes secreted from 30MV2-6 cells stimulated vascular network formation. The angiogenic activity of 30MV2-6 exosomes was compared to BM-MSC derived exosomes and tested at a dose of 200 × 10^6^ exosomes/well (Appendix A). The results showed angiogenic activity for 30MV2-6 but not BM-MSC exosomes at this dose.

We next tested whether BM-MSC exosomes were active at a higher dose. A dose of 1200 × 10^6^ exosomes/well of BM-MSC exosomes showed equivalent angiogenic activity as 30MV2-6 exosomes at 200 × 10^6^ exosomes/well, indicating a 6-fold difference in potency (Appendix A). These data demonstrated the potential for highly scalable eEPCs to produce exosomes with higher angiogenic potency than BM-MSC. Furthermore, we identified miR-126 to be the single most highly enriched miRNA in 30MV2-6 compared to BM-MSC exosomes by qPCR, showing a 50-fold enrichment of 30MV2-6 (Appendix A), further supporting the angiogenic potential of eEPC secreted exosomes. miR-126 is well established for proangiogenic activity and promotes maturation to functional vessels [54,55,56]. These data suggest an important role for enriched miR-126 in the higher angiogenic activity of eEPC-exosomes and in their potential to improve functional outcomes in the treatment of ischemic disease.

We also compared eEPC-exosomes cultured from early (32 pd) versus late passages (50 pd). The results show that the early passage eEPC exosome angiogenic activity was stable at later passage (Figure 8C). To investigate the scalability of eEPCs, we compared the angiogenic activity of eEPCs grown in monolayer culture to Quantum bioreactor (Terumo) culture after incubation in endothelial basal medium (EBM) for 72 h at 5% oxygen. The large-scale eEPC culture in the cell expansion bioreactor had equivalent angiogenic tube formation activity as conventional T-flask monolayer culture (Figure 8D). Taken together, we demonstrated stability and scalability of eEPCs, showing that large-scale expansion and late passage eEPCs maintained their angiogenic activity. This finding suggests that eEPC lines may be able to overcome a major obstacle of primary cells on the path toward producing endothelial cells at industrial scale for therapeutic applications.

## 4. Discussion

Previous studies have suggested that mesenchymal stem cells (MSCs) [4,5], endothelial progenitor cells (EPCs) [6], and cardiosphere derived cells (CDCs) [7] are attractive cell sources for vascular regeneration. However, their widespread use has been limited by their heterogeneity and lack of scalability. Endothelial cells differentiated from hPSCs (hPSC-EC) can be generated by overcoming the major bottleneck of the paucity of cells and donor heterogeneity, but these technologies have also met limitations for therapeutic usage due to heterogeneous cell populations [57]. The aim of the current study was to derive homogeneous, clonally scalable, and well-defined human endothelial cell lines. Here, we provide detailed analysis of global gene expression patterns comparing eEPC lines to adult EC (AEC) lines. The eEPC and AEC lines share a pan-endothelial specific gene expression pattern; however, the eEPC lines are distinguished from adult lines by a distinct expression of a subset of embryonic specific genes, indicating that they retain embryonic characteristics. Embryonic progenitor lines which have not crossed the embryo-fetal transition have longer replicative lifespans and may have greater regenerative properties, as we have previously proposed [21,22]. The endothelial stem cell identity of eEPC is also indicated by the expression of hematopoietic stem cells markers (CD34 and CD133) and endothelial cell markers such as CD31, KDR, vWF, Ve-Cadherin/CD144, Tie2, c-kit/CD117, and CD62E (E-selectin) [58,59] and CD45, CXCR4, CXCR2, and CCR2. Adult EPCs from different sources express different surface markers. For example, bone-marrow derived EPCs (BM-EPCs), peripheral blood-derived EPCs (PB-EPCs), and cord-blood derived EPCs (CB-EPCs) express different markers [60,61]. We found that each eEPC line has a distinct gene expression pattern, indicating that they may represent progenitors of specific endothelial cell types based on anatomical location.

In the present study, we provided comprehensive transcriptomic profiling analysis demonstrating that eEPCs differentiated from hEPs have endothelial phenotypes expressing endothelial markers such as CD31, CXCR4, KDR, and vWR, and EPC marker, CD34, as well as generating tube formation *in vitro*. Interestingly, our study showed that eEPC lines retained embryonic progenitor characteristics. We found a subset of genes that were upregulated in eEPC lines compared to AEC lines. For example, eEPC overexpressed apelin receptor (*APLNR*), insulin-like growth factor 2 (*IGF2*), the MAF BZIP transcription factor B (*MAFB*), insulin-like growth factor binding protein 2 (*IGFBP2*), non-coding RNA *H19*, carnitine palmitoyltransferase 1 C (*CPT1C*), tartrate-resistant acid phosphatase 5 (*ACP5*) and enhanced prolyl hydroxylase domain (PHD)-3 (*EGLN3*) relative to AEC. *APLNR* signaling is required for the generation of cells that give rise to HSCs [34], and there is a prominent role of the *IGF2/IGF2R* system in promoting endothelial progenitor cells homing [35]. The *MAFB* gene encodes a transcription factor that controls endothelial sprouting *in vitro* and *in vivo* [36] and plays an important role in the embryonic development of the lymphatic vascular system [62,63]. *IGFBP2* is known as a developmentally regulated gene that is highly expressed in embryonic and fetal tissues and markedly decreases after birth [64]. It promotes angiogenic and neurogenic differentiation potential of dental pulp stem cells [37]. *H19* is expressed during EC differentiation and has functional effects on EC tube formation [38]. *CPT1C* promotes human mesenchymal stem cells survival under glucose deprivation through the modulation of autophagy and is strongly associated with the epithelial-mesenchymal program [39]. *ACP5* is upregulated by transforming growth factor-β1 (*TGF-β1*) and subsequently enhances β-catenin signaling in the nucleus, which promotes the differentiation, proliferation, and migration of fibroblasts [40]. *EGLN3*, a member of the *EGLN* family of prolyl hydroxylases, increases during skeletal myoblast differentiation [65] and is associated with resistance to stress [41]. Upregulated genes in eEPCs are known to regulate endothelial function, especially during differentiation and cellular stress conditions. Further studies on the biochemical and functional properties of eEPC lines may identify novel cell types for specific therapeutic and diagnostic purposes.

We also found a subset of genes that were downregulated in eEPCs. For example, eEPC lines expressed little to no β-site APP cleavage enzyme 2 (*BACE2*), catalase (*CAT*), *CXCL1*, Interferon α-inducible protein 27 (*IFI27*), matrix Gla protein (*MGP*), and long non-coding RNA (lncRNA) maternally expressed gene 3 (*MEG3*) compared to AEC lines. The *BACE2* gene is linked to increased risk and earlier disease onset of Alzheimer’s disease. *CAT* encodes an antioxidant enzyme that plays an important role in endothelial cell pathophysiology, in shear stress response, and ultimately, in arterial aging [44]. *CXCL1* was only expressed by old osteoblasts but not young cells [66] and was known to be produced mainly by TNF-stimulated endothelial cells (ECs) as a potent chemoattractant receptor of innate immune system [45]. *IFI27* is known as an oncogene, a strong stimulator of angiogenesis, by increasing secretion of vascular endothelial growth factor (*VEGF-A*) [46]. *MGP* stimulates differentiation of endothelial cells [47], but also has been found in aortic calcified lesions of aging animal model [67]. *MEG3* negatively regulates angiogenesis and proliferation in vascular endothelial cells. *MEG3* overexpression was significantly suppressed by inhibition of miR-9 resulting in proliferation and *in vitro* angiogenesis in vascular endothelial cells [48]. Downregulated genes in eEPCs are associated with cellular organization and metabolic process. The suppression of genes associated with growth inhibition and aging is consistent with embryonic nature of the eEPCs.

Several different sources of endothelial cells have been explored for their ability to revascularize in ischemic injuries, heal wounds, or build microvasculature in engineered tissues. Human umbilical vein endothelial cells (HUVECs) have been a robust source of ECs with proven capability of forming stable microvessels, which are capable of capillary morphogenesis compared to hPSC-ECs [57]. From our comprehensive transcriptomic analysis comparing 13 eEPC lines to 15 AEC lines, we observed that HUVECs expressed a distinct gene expression pattern distinguished from the 14 adult EC lines consistent with their early newborn/fetal nature (Figure 5 and Appendix A). Indeed, a subset of genes was similarly expressed both in HUVEC and the 13 eEPC lines. These included *IFITM1*, *NNAT*, *H19*, *PLVAP*, and *IGF2*, which were highly upregulated and *CYTL1*, *KRT7*, *CXCL1*, and *CXCL8*, which were downregulated in HUVECs and eEPC lines, but not the AEC lines (Appendix A). Our findings comparing eEPC gene expression profiles to HUVEC are consistent with HUVECs representing a cell state that is in between embryonic and adult. The increased regenerative capacity of HUVEC compared to AEC is consistent with our somatic restriction concept that cells lose regenerative capacity in stages starting with the embryo to fetal transition and continuing through the fetal to newborn and adult transitions [22]. More studies are needed to compare the regenerative capacity of eEPC, HUVEC and AEC lines.

Derivation of eEPC lines that are homogeneous clonal cells with batch-to-batch consistency in production provides a unique competitive advantage for unparalleled industrial scale production that is needed for therapeutic application in various ischemic diseases. The overall advantages of hEP lines are (1) identity and homogeneity, (2) scalability and stability, and (3) diversity (allowing selection of optimal production and high potency.

There are several groups establishing adult (non-embryonic) stem cells such as endothelial progenitor cells (EPCs), MSCs, and CDCs for clinical application in the treatment of ischemic disease. However, the source of these primary cell lines results in cellular heterogeneity both within a donor line and between donor lines. Thus, there is a manufacturing bottleneck for the consistent production of therapeutic ECs on an industrial scale needed to treat a large cardiovascular disease patient population. We addressed this issue by developing a bank of embryonic progenitor cell lines derived by early-stage clonal isolation from partially differentiated hPS cells [21]. This method has resulted in a diverse collection of distinct cell lines with increased homogeneity, stability, and scalability because of their clonal purity and long embryonic telomere length [21]. The resulting lines feature homeobox gene expression consistent with an endoderm, ectoderm, mesoderm, or neural crest origin [26]. We have identified a variety of cell fates including bone, cartilage, smooth muscle cells, pericytes, endothelial cells, white and brown adipose tissue, with various applications in regenerative medicine [21,24,25,26]. The diversity that we have generated in our hEP line library allows us to select for cell lines that are tailored to specific applications. We can therefore identify candidate lines based on cell line stability and scalability in addition to regenerative potency. We used two different directed differentiation variations on our previous method [21,24,25,26,68], to generate 46 candidate eEPC lines from which we validated 13 distinct endothelial lines. Interestingly, our endoderm differentiation method led to successful endothelial cell line isolation. An endodermal origin of endothelial cell has recently been reported [69,70]. Taken together, these results demonstrate that directed differentiation from hPS cells and clonal selection can be used to isolate diverse, scalable endothelial progenitor lines. The resulting cell lines have the potential for the development of cell and exosome-based therapeutics.

## 5. Conclusions

Here, we demonstrate the ability to derive clonally pure endothelial cells (eEPCs) from hPS cells using our two-step process. We derived 13 eEPC lines and characterized them as endothelial by transcriptomic analysis, biochemical markers, and vascular function. Further, we have shown early evidence of the angiogenic potential of exosomes secreted from eEPC lines. We are currently developing eEPC exosomes manufacturing methods and investigating the potential therapeutic uses of eEPC exosomes for treating ischemic conditions like stroke by improving vascular and neurological recovery. Our findings suggest that eEPC lines will be a valuable source of both cells and exosomes for use in therapeutic vascular regeneration.

## 6. Patents

Exosomes from Clonal Progenitor Cells, 2022 (US 11274281); Exosomes from Clonal Progenitor Cells, 2019 (US 10,240,127); Methods and compositions for targeting progenitor cell lines, 2019 (US 10,227,561).

## Figures and Tables

**Figure 1 biomedicines-11-02777-f001:**
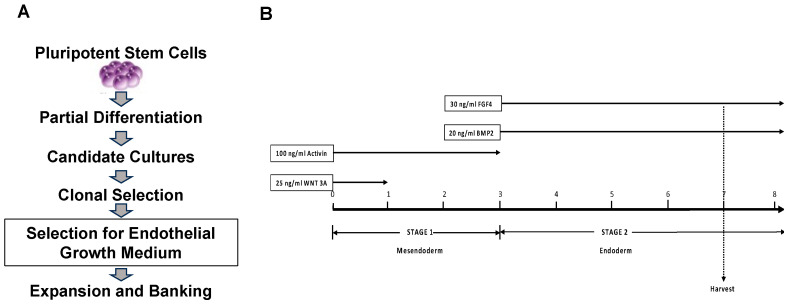
Derivation of Endothelial Progenitor Cell Lines, (**A**) Schematic illustration of establishing embryonic endothelial progenitor cell (eEPC) lines from clonal selection and (**B**) eEPC differentiation strategy from hPSCs, Protocol “30”.

**Figure 2 biomedicines-11-02777-f002:**
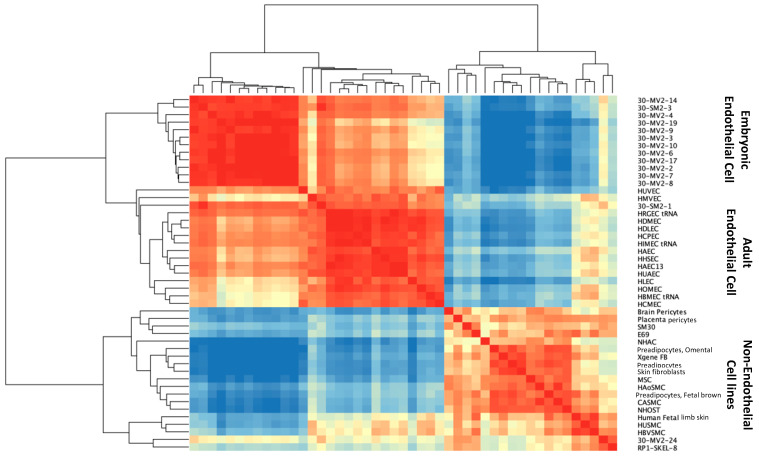
Pearson correlation analysis of 14 human embryonic endothelial progenitor lines (eEPC), 15 adult endothelial cell (AEC) lines and 17 non-endothelial cell lines. Color ranged as high expression (red) to low expression (blue), p (Corr) cut-off = 0.05.

**Figure 3 biomedicines-11-02777-f003:**
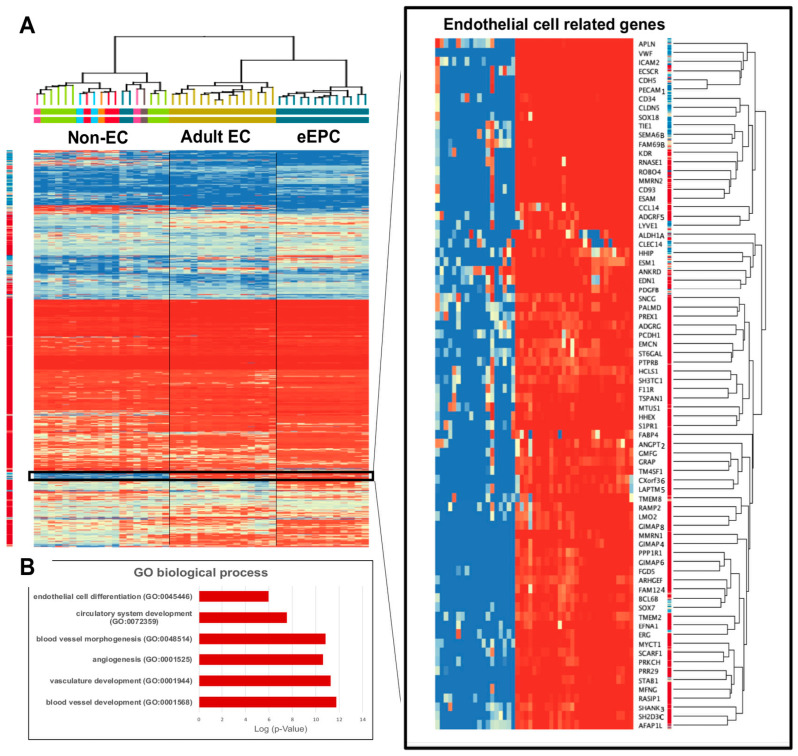
Comparision of endothelial characteristics of 14 clonal eEPC lines to 15 human adult endothelial cell (AEC) lines and 17 non-endothelial cell lines, (**A**) Transcriptomic RNA-seq hierarchical analysis showed that 13 out of 14 eEPC lines shared a group of endothelial specific genes with primary AEC lines, moderate *t*-test and p (Corr) cut-off = 0.05, Color ranged as high expression (red) to low expression (blue), (**B**) Biological gene ontology (GO).

**Figure 4 biomedicines-11-02777-f004:**
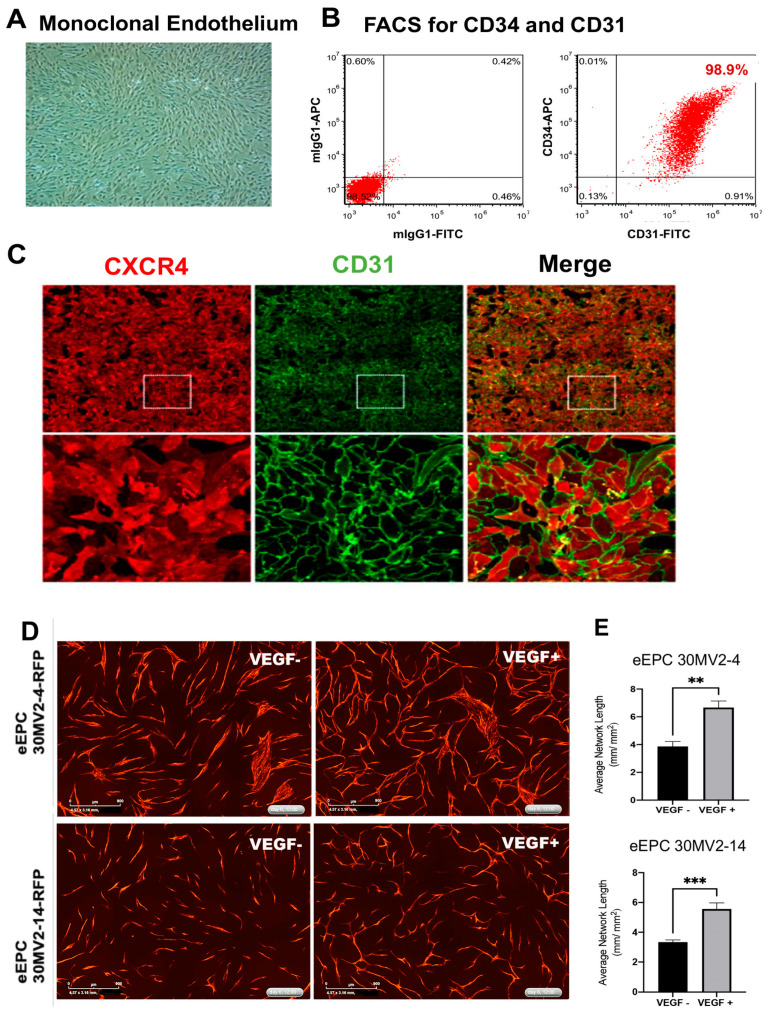
Characterization of clonal endothelial embryonic progenitor cell lines, (**A**) Phase contrast image of the representative eEPC line, 30MV2-6, showing uniform cobblestone morphology, (**B**) Flow cytometric analysis of endothelial markers, CD34 and CD31 with IgG1 negative control, (**C**) Immunofluorescence staining of endothelial markers, CXCR4 and CD31, Images in lower panel show the enlarged images in the white box, (**D**) Live co-culture angiogenesis assay images at day 6 showing RFP-labeled eEPCs (30MV2-4 and 30MV2-14) tube formation in response to VEGF (related live tube formation video shown in Appendix A). (**E**) Bar graphs show quantitative increase in tube network length in response to VEGF in both cell lines (** *p* < 0.01, *** *p* < 0.001).

**Figure 5 biomedicines-11-02777-f005:**
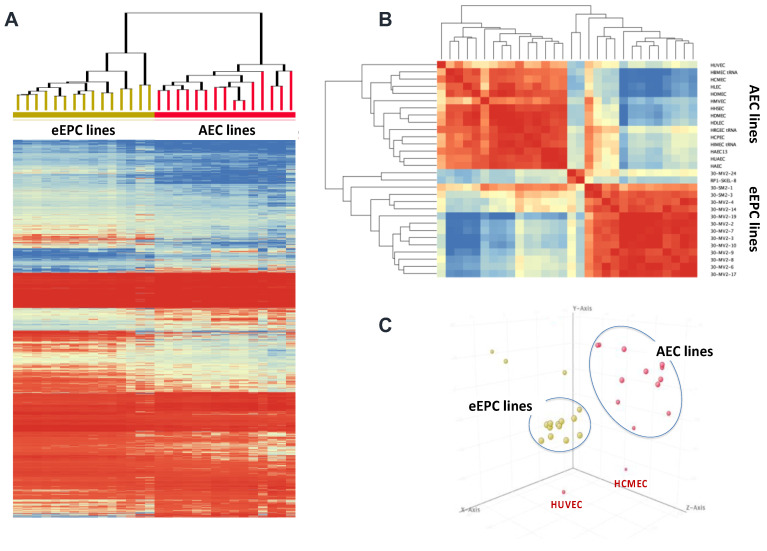
RNAseq Transcriptomic analysis showing retention of embryonic phenotype in 14 eEPC lines compared to 15 adult endothelial cell (AEC) lines including HUVEC (Human Umbilical Vein Endothelial Cell) and HCMEC (Human Cardiac Microvascular Endothelial). (**A**) RNAseq cluster analysis heatmap using moderate *t*-test, (**B**) Gene Correlation analysis, p (Corr) cut-off = 0.05, Color ranged as high expression (red) to low expression (blue), and (**C**) principal component analysis (PCA), yellow dots represent each eEPC lines and red dots represent each AEC lines.

**Figure 6 biomedicines-11-02777-f006:**
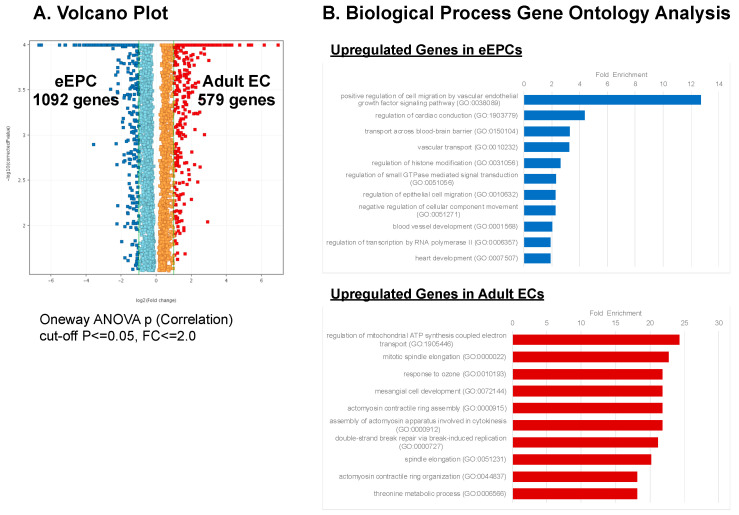
Differential gene expression analysis of Adult EC lines vs. eEPC lines. (**A**) Volcano Plot using One-way ANOVA p (Correlation), cut-off *p* ≤ 0.05, Fold Change ≤ 2.0 and (**B**) Gene Ontology Pathway enrichment analysis in eEPC-enriched genes and AEC-enriched genes.

**Figure 7 biomedicines-11-02777-f007:**
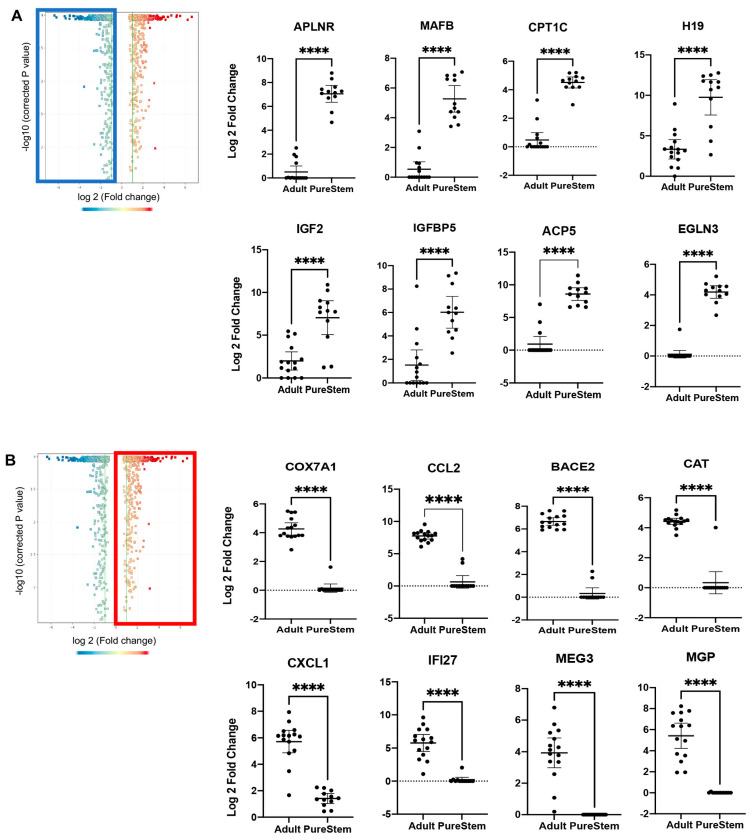
Gene expression comparison of 13 eEPC lines and 15 adult EC (AEC) lines, (**A**) Highly expressing genes expression in eEPC lines compared to adult EC (AEC) lines and (**B**) Highly expressing genes expression in AEC lines compared to eEPC lines, Volcano Plot was used one-way ANOVA, p (Correlation), cut-off *p* ≤ 0.05, FC ≤ 2.0. Statistical analysis between two groups (13 eEPC lines vs. 15 AEC lines) was performed using the unpaired student’s *t*-test with two-tailed *p* value (**** *p* < 0.0001).

**Figure 8 biomedicines-11-02777-f008:**
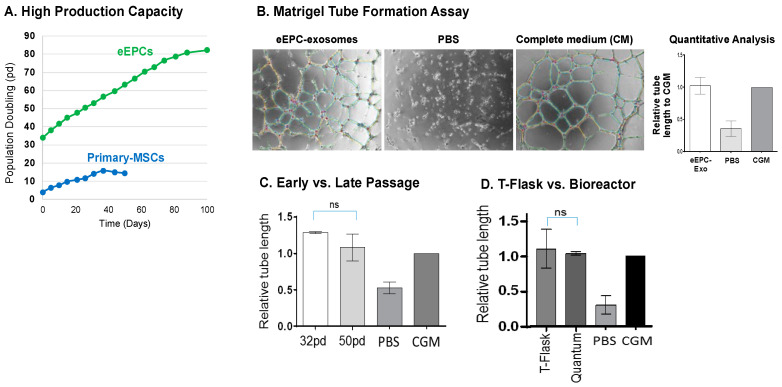
Stable and scalable production of embryonic endothelial progenitor cells (eEPCs), (**A**) Cumulative population doublings of eEPC line (30MV2-6) was up to 80 population doublings, compared to primary MSCs, which began to senesce and lose proliferative capacity after 8 to 12 population doublings, (**B**) Angiogenic activity of eEPC (30MV2-6) secreted exosomes in HUVEC vascular network tube forming assay was equivalent to CGM (Complete Growth Medium containing VEGF, FGF, and IGF). Relative tube length = sample total tube length/CGM total tube length, (**C**) Angiogenic activity measured by HUVEC tube forming assay was retained from early to late passage (30 pd to 50 pd), and (**D**) Exosomes from eEPC grown in a Quantum cell expansion bioreactor (Terumo BCT) resulted in no loss of angiogenic activity compared to T-flask culture. The negative control is vehicle (PBS) in BM (serum-free basal medium). Error bars indicate the standard deviation from the mean of technical (n = 3) replicates, representative of at least 2 biological repeats. ns represents statistically not significant.

## Data Availability

Data is available upon request.

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
