# Peer review of "Clonal and Scalable Endothelial Progenitor Cell Lines from Human Pluripotent Stem Cells"

_biomedicines, 2023, doi:10.3390/biomedicines11102777_

Round 1
Reviewer 1 Report
The aim of this manuscript is to explore the feasibility of inducing human embryonic stem cells into embryonic endothelial progenitor cells as a potential therapeutic approach for ischemic heart disease.
- We kindly request a high-resolution image for Figure 4.
- In the legend for Figure 5, please consider providing clearer abbreviations for better comprehension, such as AEC lines, eEPC line, HUVEC, and HCMEC.
- In Figure 7A, it would be beneficial to include p-values for comparison, particularly for H19, IGF2, and IGFBP5. The bar graph indicates evident overlap between the two groups.
- Regarding Figure 7A and the "H19 and IGF2 gene" expression, it may be helpful to set the expression levels for adults at "0" to provide a clearer reference point.
- Similarly, for Figure 7B and the CXCL1 gene
Reviewer 2 Report
The work by Lee et al focuses on the generation and characterization of embryonic endothelial progenitor cells (eEPCs) as a source of therapeutic cells for many different pathological conditions that are either caused or affected by impaired by vascular deficiencies.
The transcriptomic data and its analysis is robust, and they strongly suggest a similarity between the eEPCs generated and adult endothelial cell populations.
However, more robust functional data around the physiological behavior of the eEPCs is warranted. For example, assays focused on leakiness of the tubes formed, as well as barrier function and monolayer integrity of cells in standard 2D monolayers would indicate whether those transcriptional similarities translate into functional similarities, making the eEPCs as desirable as a cell therapy as the authors correctly propose.
Minor comments
Table 1 - Although the information on the optimal media composition is important and relevant, this reviewer finds it would be best suited in either the materials and methods section, or as supplementary information.
Figure 4B - The authors should provide data obtained from a non-endothelial cell of choice analyzed using the same flow cytometry markers and protocol as negative control, and compare the 30MV2-6 against those cells
Figure 8B and C- Rather than just providing the graphed quantification data, the authors should also provide a representative image of the tubes formed.
Lines 438-441 seem copied and pasted from author instructions. They should be removed.
Round 2
Reviewer 2 Report
The authors satisfactorily addressed this reviewers' comments and the manuscript is now better suited for publication. No further comments.